# Synergistic Anti-Inflammatory Effects of Lipophilic Grape Seed Proanthocyanidin and Camellia Oil Combination in LPS-Stimulated RAW264.7 Cells

**DOI:** 10.3390/antiox11020289

**Published:** 2022-01-31

**Authors:** Linli Zhang, Juan Chen, Ruihong Liang, Chengmei Liu, Mingshun Chen, Jun Chen

**Affiliations:** 1State Key Laboratory of Food Science and Technology, Nanchang University, Nanchang 330047, China; 357900210032@email.ncu.edu.cn (L.Z.); liangruihong@ncu.edu.cn (R.L.); liuchengmei@ncu.edu.cn (C.L.); chen-jun@ncu.edu.cn (J.C.); 2Moutai Institute, Renhuai 564501, China; kilooy@163.com

**Keywords:** Lipophilic grape seed proanthocyanidin, camellia oil, anti-inflammatory, synergistic effect, RAW264.7 cells

## Abstract

Combination drug therapy has become an effective strategy to control inflammation. Lipophilic grape seed proanthocyanidin (LGSP) and camellia oil (CO) have been independently investigated to show anti-inflammatory effects, but their synergistic anti-inflammatory effects are unknown. The aim of this study was to investigate the synergistic anti-inflammatory effects of LGSP and CO. The anti-inflammatory activity of LGSP and CO individual or in combination on RAW264.7 cells was detected by MTT assay, Griess reagent, RT-PCR, 2′,7′-dichlorfluoroescein diacetate and Western blot analysis. The combined treatment of LGSP with CO (20 μg/mL and 1 mg/mL) synergistically suppressed the production of NO, TNF-α, IL-6 and ROS. Further studies showed that the synergistic effect was attributed to their suppression of the activation of NF-κB and MAPK signaling pathways. Overall, our findings demonstrate the potential synergistic effect between LGSP and CO in LPS-induced inflammation.

## 1. Introduction

Inflammation is an intricate defensive response of the body to achieve self-protection against various irritation and infection. Excess inflammation is a favorable condition for the formation of various chronic diseases, including diabetes, colitis, obesity and even cancer [1,2]. NO is an inflammatory mediator, and excess NO in immune-inflammatory diseases can induce the production of some pro-inflammatory factors, such as TNF-α, IL-6 and IL-1β, which are the main indicators of the severity of inflammation in vivo [3]. IL-10 is a cytokine with strong anti-inflammatory properties that play an important role in the immune response against pathogens [4]. With the rapid pace of modern life and the deterioration of the living environment, people are easily exposed to the stimulation factors, such as stress, staying up late, unhealthy diet and bacteria, leading to inflammation [5,6]. Nonsteroidal anti-inflammatory drugs and corticosteroids are the most commonly used anti-inflammatory drugs, but they have some side effects [7,8]. Thus, it is necessary to develop novel and more effective anti-inflammatory drugs continuously.

Phytochemicals from various foods such as fruits, vegetables, grains, nuts and cocoa/chocolate, including phenols, terpenoids and organic sulfur compounds, have been reported to have anti-inflammatory activity [9,10]. Grape seed proanthocyanidin (GSP), a natural antioxidant of polyphenols in grape seed, was proven to have anti-inflammatory activity [11]. Lipophilic grape seed proanthocyanidins (LGSP), synthesized by enzymatic esterification of GSP, showed stronger antioxidant [12], anticancer [13] and anti-inflammatory activities [14], compared with GSP. Based on the complexity of digestion, absorption, metabolism and interaction between bioactive phytochemicals and food, combined drug therapy has become an effective strategy to control inflammation, which can improve pharmacological activity and decrease side effects by acting on multiple targets of inflammation. Camellia oil (CO), edible oil with extremely high nutritional value, contains abundant unsaturated fatty acids and phytochemicals, including squalene, sterols, tocopherols and polyphenols [15], exhibiting a variety of bioactivities, such as antioxidant, antitumor and anti-inflammatory [16]. Pallares et al. [17] reported that proanthocyanidins and polyunsaturated fatty acids showed synergistic anti-inflammatory effects in vitro. Thus, we hypothesize that the combination of LGSP and CO might amplify their anti-inflammatory effect.

This study aimed to investigate whether the combination of LGSP and CO had a synergistic anti-inflammatory effect. The underlying mechanism of the synergistic anti-inflammatory effect was also explored. The results might provide the basis for the combined utilization of LGSP and CO as an effective anti-inflammatory agent in the future.

## 2. Materials and Methods

### 2.1. Materials and Reagents

CO was obtained from Jiangxi Qiyunshan Food Co., Ltd. (Ganzhou, China). DMEM, penicillin/streptomycin and FBS were purchased from Gibco (Grand Island, NY, USA). Lipopolysaccharides (LPS) and MTT were provided by Sigma-Aldrich (St. Louis, MO, USA). Griess reagent NO assay kit and BCA kit were purchased from Beyotime Biotechnology Co. (Shanghai, China). The antibodies were purchased from Cell Signaling Technology (Beverly, MA, USA). Trizol reagent was purchased from Tiangen Biochemical Technology Co. (Beijing, China).

### 2.2. Preparation of LGSP

LGSP was prepared as described in our previous literature [12]. GSP and lauric acid were added into a screw-capped glass bottle containing ethanol solvent, using Lipozyme TLIM as the catalyst. The reaction was carried out at 45 °C for 12 h and stopped by removing the enzyme. LGSP was obtained by concentrating the product (Appendix A).

### 2.3. Cell Culture

RAW264.7 cells were purchased from ATCC (Rockville, MD, USA) and cultured in DMEM supplemented with 10% FBS at 37 °C under 5% CO_2._

### 2.4. Cell Viability Assay

The cytotoxicity of LGSP and CO was determined by MTT assay [18]. Cells (1 × 10^4^ cells/mL) were seeded on 96 well-plates. After adherence, cells were treated with LGSP (5–80 μg/mL) and/or CO (0.125–2 mg/mL) for 24 h. An amount of 200 μL of MTT reagent (0.5 mg/mL) was added and incubated for 4 h. Then, DMSO was added and measured at 490 nm by a microplate reader (KHB ST-360, Shanghai, China).

### 2.5. NO Production Measurement

NO production was detected by Griess reagent [19]. Cells (1 × 10^5^ cells/mL) were seeded into 96 well-plates. After attaching, cells were pretreated with different concentrations of LGSP and/or CO for 2 h, and then the LPS (1 μg/mL) was added and co-treated for 24 h. The supernatant was then mixed with Griess reagent in the ratio of 1:1 and incubated for 10 min in the dark. The absorbance was measured at 540 nm by a microplate reader (KHB ST-360, Shanghai, China).

### 2.6. Synergistic Effect Analysis

The synergistic effect of LGSP and CO on the NO level was analyzed by CompuSyn software 2.0, as described in the literature [20]. The NO inhibition data collected after treatment with LGSP and/or CO were entered into the CompuSyn software [21].

### 2.7. Quantitative Real-Time PCR Analysis

The Trizol reagent was used to extract the total cellular RNA, and the concentration of RNA was measured using NanoDrop 1000 Spectrophotometer (DeNovix). An amount of 2 μg of total RNA was converted to single-stranded cDNA from each sample, which was then amplified by Brilliant II SYBR Green QRT-PCR Master Mix Kit. The gene expression was quantitatively detected by the CFX96 Real-Time PCR Detection System (Bio-Rad, Hercules, CA, USA). The primer pairs were synthesized by Integrated DNA Technologies, Inc. (Coralville, IA, USA), and the following sense and antisense primer sequences used for RT-PCR analysis were: TNF-α, 5′-CACCACCATCAAGGACTCAAAT-3′ (forward) and 5′-CAGGGAAGAATCTGGAAAGGT-3′ (reverse); IL-6, 5′-CTGGGAAATCGTGGAAATGAG-3′ (forward) and 5′-GACTCTGGCTTGTCTTTCTTGTTA-3′ (reverse); IL-1β, 5′-AGATAGAAGTCAAGAGCAAAGTGGA-3′ (forward) and 5′-TGGGGAAGGCATTAGAAACAG-3′ (reverse); IL-10, 5′-TGGACAACATACTGCTAACCGAC-3′ (forward) and 5’-ATGCTCCTTGATTTCTGGGC-3′ (reverse); GAPDH, 5′-AGGTCGGTGTGAACGGATTTG-3′ (forward) and 5′-TGTAGACCATGTAGTTGAGGTCA-3′ (reverse). The 2^−ΔΔCt^ method was used to calculate the copy number of each transcript relative to the GADPH [22].

### 2.8. Measurement of Reactive Oxygen Species (ROS) Production

ROS were detected using 2′,7′-dichlorfluoroescein diacetate (DCFHDA) [23]. Cells (1 × 10^5^ cells/well) were seeded into 6-well plates for 12 h. After the adherent, cells were pretreated with various concentrations of LGSP and/or CO for 2 h and then exposed to LPS (1 μg/mL) for 12 h. The cells were then treated with 10 μM DCFH-DA for 30 min in the dark and detected by a fluorescence microplate reader (Nikon Eclipse Ti, Tokyo, Japan).

### 2.9. Western Blot Analysis

Cells (1 × 10^5^ cells/well) were plated into 6-well plates. After incubated for 12 h, cells were pretreated with various concentrations of LGSP and/or CO for 2 h, and LPS (1 µg/mL) was subsequently added and incubated for 6 h. The protein was obtained by lysing cells and quantified using the BCA protein analysis kit. Cell lysates were denatured in SDS loading buffer, separated by SDS-PAGE gel and transferred onto immunoblot membranes. After being blocked at room temperature for 1 h, the membranes were incubated at 4 °C overnight with a specific primary antibody, followed by a peroxidase-conjugated secondary antibody at room temperature for 1 h. Protein bands were detected using the ChemiDoc XRS+ system (Bio-Rad).

### 2.10. Statistical Analysis

All experiments were performed at least in triplicate and results were presented as mean ±standard deviation. Significant differences between means were determined using Duncan’s multiple range test (*p* < 0.05).

## 3. Results and Discussion

### 3.1. Effect of LGSP and CO on Cell Viability of RAW264.7 Cells

The effect of LGSP and CO on RAW264.7 cell viability was detected by MTT assay to eliminate the influence of cytotoxicity on their anti-inflammatory activity. As shown in Figure 1, LGSP at doses of 5–80 μg/mL and CO at doses of 0.125–2 mg/mL had no effect on the viability of RAW264.7 cells. Meanwhile, the combination of LGSP and CO in the ratio of 1:25, 1:50, 1:100 (LGSP: CO) also had no effect on the viability of RAW264.7 cells. The results suggested that LGSP and/or CO were not cytotoxic to RAW264.7 cells at the above doses.

### 3.2. Effect of LGSP and CO on Inhibition of NO Production

NO is an endogenous gaseous signal molecule, which can be produced as an inflammatory mediator in LPS-induced macrophages [24]. The effect of LGSP and CO on NO levels in LPS-induced RAW264.7 cells was displayed in Figure 2. LPS stimulation led to a sharp increase in NO production (*p* < 0.05). LGSP inhibited LPS-induced NO in a dose-dependent manner by 2.43, 4.71, 12.03, 20.17 and 29.16% at doses of 5, 10, 20, 40 and 80 μg/mL, respectively (Figure 2A). While treatment with CO resulted in a dose-dependent inhibition on NO production by 2.19, 11.22, 20.49, 21.47 and 37.41% at 0.125, 0.25, 0.5, 1 and 2 mg/mL, respectively (Figure 2B). Notably, the combination of LGSP and CO showed a stronger inhibitory effect on NO production than LGSP or CO individual (*p* < 0.05), and the inhibitory effect depended on the concentration and ratio of LGSP and CO (Figure 2C). The inhibition rate of each concentration gradient with LGSP and CO ratio of 1:50 was higher than that of 1:25 and 1:100. Notably, treatment with LGSP (40 μg/mL) and CO (2 mg/mL) caused a 20.17% and 37.41% reduction in NO production, respectively, but the combined treatment with LGSP and CO resulted in a 53.94% inhibition rate of NO production.

### 3.3. Synergistic Effect Analysis

Based on the results of LGSP and CO treatment on NO production, the mode of interaction between LGSP and CO in inhibiting NO production was further determined. Figure 3 showed the isobologram analysis for the combination of LGSP and CO. As shown in Figure 3A–C, the combination of LGSP and CO in the ratio of 1:50 to achieve 90% inhibition, 75% inhibition and 50% inhibition were all below the corresponding lines, suggesting that there was a synergistic anti-inflammatory effect between LGSP and CO.

The combination index (CI), a quantitative indicator of the extent of drug interaction, is used to determine the presence of synergy or antagonism between drugs. A lower CI value represents a stronger synergistic effect. Figure 3D displayed that CI values of LGSP and CO (ratio of 1:50) were between 0.76 and 0.84. LGSP (20 μg/mL) combined with CO (1 mg/mL) exhibited the lowest CI value (0.76), indicating the best synergistic effect. Therefore, the combination of LGSP (20 μg/mL) and CO (1 mg/mL) was selected for further study.

### 3.4. Effect of LGSP and CO on the mRNA Expression of Inflammatory Cytokines

In inflammatory diseases, pro-inflammatory cytokines are over-expressed in LPS-induced RAW264.7 cells, and these inflammatory cytokines were verified as the major indicators of the severity of inflammation in vivo [25,26]. Then, the influence of LGSP and CO on the expression of TNF-α, IL-6 and IL-1β mRNA was detected by RT-PCR. As shown in Figure 4, LPS dramatically enhanced the expression of TNF-α, IL-6, IL-1β and IL-10 mRNA, compared with control (*p* < 0.05). LGSP treatment inhibited the expression of TNF-α and IL-6 mRNA by 36.8% and 14.6%, which was in line with our previous study [14]. CO caused 42.4% and 46.08% inhibition of TNF-α and IL-6 mRNA. Cheng et al. [27] reported that CO could decrease the level of IL-6 in the rat intestinal mucosal injury model. The combination of LGSP and CO strongly inhibited TNF-α and IL-6 mRNA levels by 90.5% and 74.8%, respectively (Figure 4A,B). However, no significant difference was observed in the mRNA expression of IL-1β and IL-10 between their combination treatment and LGSP or CO individual treatment (Figure 4C,D). These results clearly demonstrated that LGSP combined with CO had a synergistic higher inhibitory effect on TNF-α and IL-6 mRNA expression than LGSP or CO individual.

### 3.5. Effect of LGSP and CO on LPS-Induced ROS

Macrophages release ROS after being stimulated, leading to cell and tissue damage, thereby further stimulating the inflammatory condition [28]. ROS is one of the major factors that cause the enhancement of oxidative stress, which plays a vital role in the pathogenesis of chronic inflammation [29]. Thus, the effect of LGSP (20 μg/mL) and CO (1 mg/mL) on LPS-induced ROS was evaluated. As shown in Figure 5, LPS stimulation resulted in a remarkable increase in the fluorescence intensity of ROS in RAW264.7 cells, compared with control (*p* < 0.05). This intensity was remarkably decreased after treatment with LGSP and/or CO. The combination of LGSP and CO exhibited a stronger inhibitory effect (93.5%) than LGSP (38.3%) or CO (58.6%) individually on LPS-induced ROS. Bumrungpert et al. [30] reported that the CO-enriched diet could decrease the biomarkers of oxidative stress caused by ROS in hypercholesterolemia. Combined with our experimental data, the synergistic anti-inflammatory activity of LGSP and CO might be attributed to their ability to inhibit oxidative stress-induced damage by scavenging ROS, at least partly.

### 3.6. Effect of LGSP and CO on LPS-Induced iNOS, NF-κB and MAPK Pathway

In order to further explore the potential synergistic anti-inflammatory mechanism of LGSP and CO, iNOS and two typical inflammatory pathways, NF-κB and MAPK, were detected in LPS-stimulated RAW264.7 cells. iNOS can be activated by endotoxin and pro-inflammatory mediators in macrophages and regulate the production of NO. As shown in Figure 6A, LGSP at 20 μg/mL and CO at 1 mg/mL significantly decreased iNOS levels compared to the LPS-treated group. However, no significant difference in iNOS expression was found between LGSP or CO individual treatment and their combination treatment. NO regulation is a complex and intensive process. When cells are stimulated by cytokines or immune microorganisms, iNOS catalyzes the production of a large number of NO, and the variation in NO content in cells is affected by substrate availability, iNOS protease activity, expression level or other subtypes of NOS enzyme activity [31]. The synergistic inhibitory effect of LGSP and CO on NO was not entirely determined by their effect on iNOS activity.

As a nuclear transcriptional factor, NF-κB regulates the expression of various inflammation and immune response-related genes [32]. It can be activated in LPS-stimulated macrophages, followed by the phosphorylation of the repressor protein IκBα and dissociated from NF-κB, leading to NF-κB translocation into the nucleus to induce the expression of inflammatory cytokines genes [33]. In order to clarify whether LGSP and CO could regulate NF-κB translocation, the nuclear and cytoplasmic fractions were determined by Western blotting analysis. As shown in Figure 6B, LPS treatment significantly promoted the nuclear translocation of NF-κB (p65), and the IκBα was obviously phosphorylated, compared with control (*p* < 0.05). However, treatment with LGSP and CO individual or in combination (20 μg/mL and 1 mg/mL) could attenuate the activation of NF-κB induced by LPS. Particularly, the inhibitory effect of LGSP combined with CO on nuclear translocation of NF-κB (p65) and phosphorylation of IκBα was markedly stronger than that of LGSP and CO individuals. Sanchez-Fidalgo et al. [34] reported that squalene prevented the continuous increase in inflammatory cytokine levels by inhibiting NF-κB in DSS-induced acute colitis. Meanwhile, tea polyphenol could inhibit the release of pro-inflammatory cytokines and the activation of NF-κB in macrophages [35]. Squalene and tea polyphenols are the main active phytochemicals in CO; Thus, LGSP combined with CO synergistically inhibited the activation of NF-κB, which may be attributed to the complementary and overlapping effects between LGSP and these active substances in CO.

Additionally, we examined the effects of LGSP and CO on LPS-stimulated MAPK by monitoring the expression of p-p38, p-JNK and p-ERK. MAPK plays a critical role in the initiation of the inflammatory response by participating in the regulation of the synthesis of inflammatory cytokines at the levels of transcription and translation [36]. LPS treatment resulted in excessive expression of p-p38, p-JNK and p-ERK (Figure 6C). Treatment with LGSP and CO individuals obviously suppressed the expression of p-p38, p-JNK and p-ERK, while treated with LGSP and CO in combination (20 μg/mL and 1 mg/mL) caused a stronger inhibition. Strikingly, the combination of LGSP and CO displayed the most remarkable synergistic effect on the down-regulation of p-JNK. Eriocitrin combined with resveratrol could strongly inhibit the expression of NO, TNF-α and IL-1β by reducing the levels of p-P38 and p-JNK in LPS-induced RAW264.7 cells, but p-ERK was not affected [37]. Tocopherols, a rich phytochemical in CO, was shown to reduce ERK and p38 phosphorylation in the inflammatory response of bronchial and alveolar epithelial cells [38]. In addition, LGSP showed a stronger inhibitory effect on p-JNK than GSP in our previous study [14]. The results indicated that the combination of LGSP and CO (20 μg/mL and 1 mg/mL) could inhibit the pro-inflammatory mediators (NO, TNF-α and IL-6) and the ROS mediated through the suppression of NF-κB and MAPK signaling pathways (Figure 7).

## 4. Conclusions

The present study demonstrated for the first time that LGSP and CO possessed synergistically anti-inflammatory effects. LGSP and CO in combination suppressed the production of NO, TNF-α and IL-6 and ROS in LPS-stimulated RAW264.7 cells by blocking NF-κB and MAPK signaling pathways. These results may provide a novel and effective agent for the treatment of inflammatory.

## Figures and Tables

**Figure 1 antioxidants-11-00289-f001:**
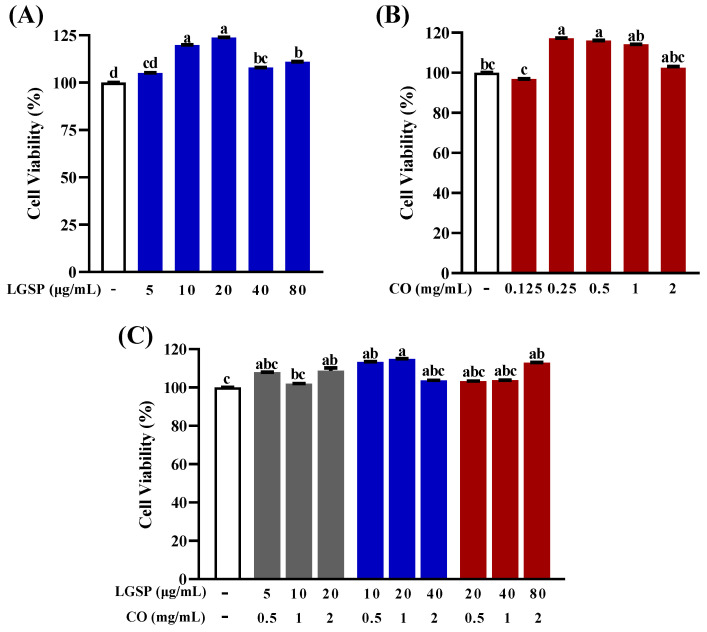
Effect of LGSP (**A**), CO (**B**) and their combination (**C**) on viability of RAW264.7 cells. All values are means ± SD, *n* = 3. Different letters above the bars indicate significant differences (*p* < 0.05).

**Figure 2 antioxidants-11-00289-f002:**
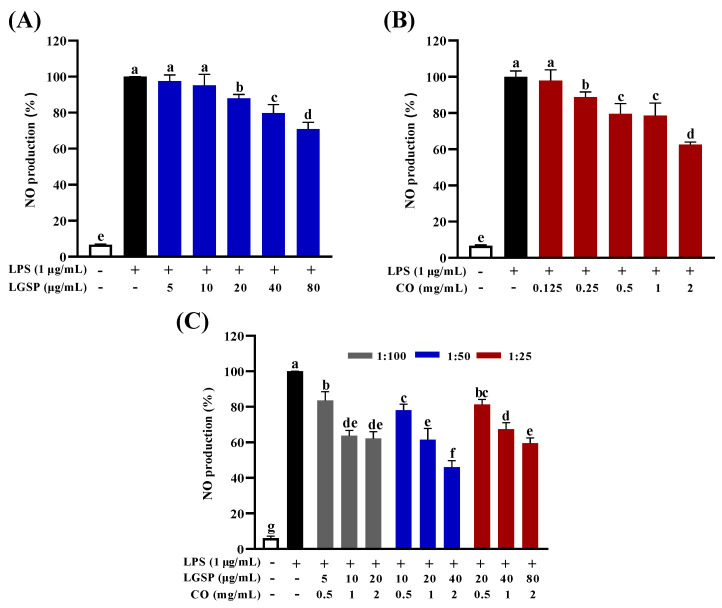
Effect of LGSP (**A**), CO (**B**) and their combination (**C**) on NO production in RAW264.7 cells. All values are means ± SD, *n* = 3. Different letters above the bars indicate significant differences (*p* < 0.05).

**Figure 3 antioxidants-11-00289-f003:**
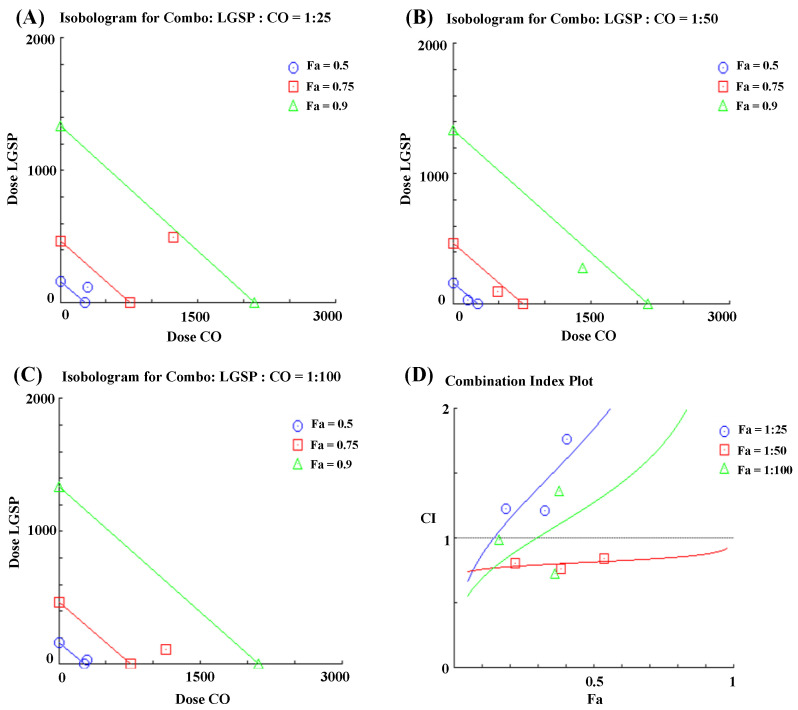
Isobologram curves for the combination of LGSP and CO in ratios of 1:25 (**A**), 1:50 (**B**) and 1:100 (**C**) on inhibition of NO production. Combination index (CI) for LGSP and CO on inhibition of NO production in LPS-stimulated RAW264.7 cells (**D**).

**Figure 4 antioxidants-11-00289-f004:**
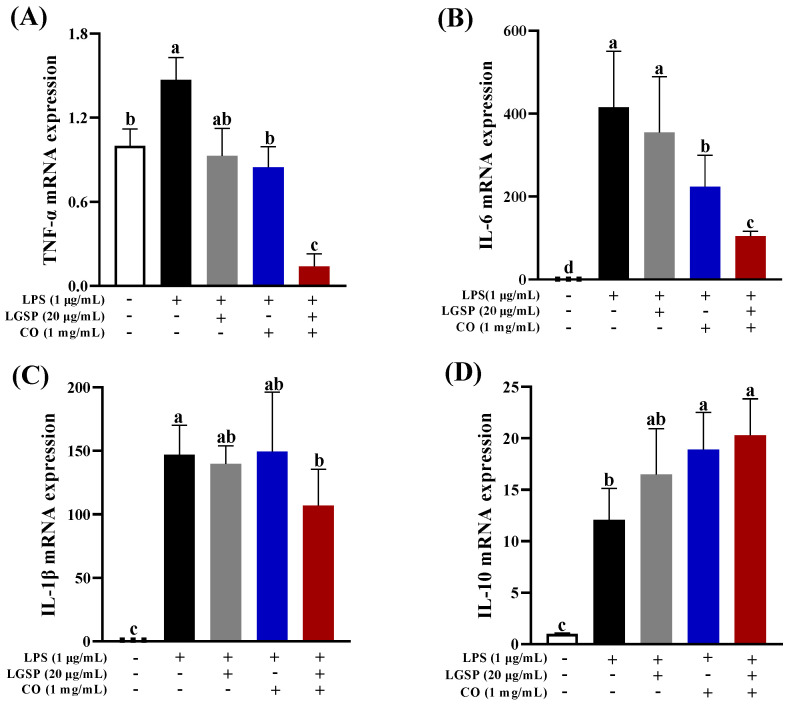
Effects of LGSP, CO and their combination on LPS-induced production of inflammatory cytokines TNF-α (**A**), IL-6 (**B**), IL-1β (**C**) and IL-10 (**D**) in RAW264.7 cells. All values are means ± SD, *n* = 3. Different letters above the bars indicate significant differences (*p* < 0.05).

**Figure 5 antioxidants-11-00289-f005:**
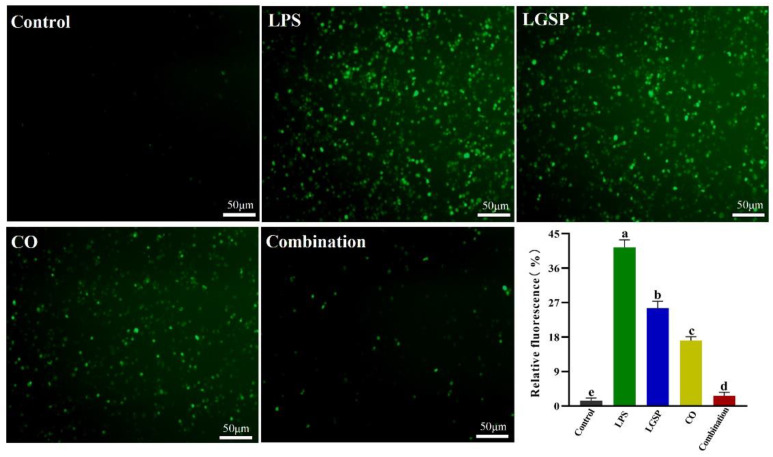
Effect of LGSP (20 μg/mL), CO (1 mg/mL) and their combination on ROS in LPS-induced RAW264.7 cells. All values are means ± SD, *n* = 3. Different letters above the bars indicate significant differences (*p* < 0.05).

**Figure 6 antioxidants-11-00289-f006:**
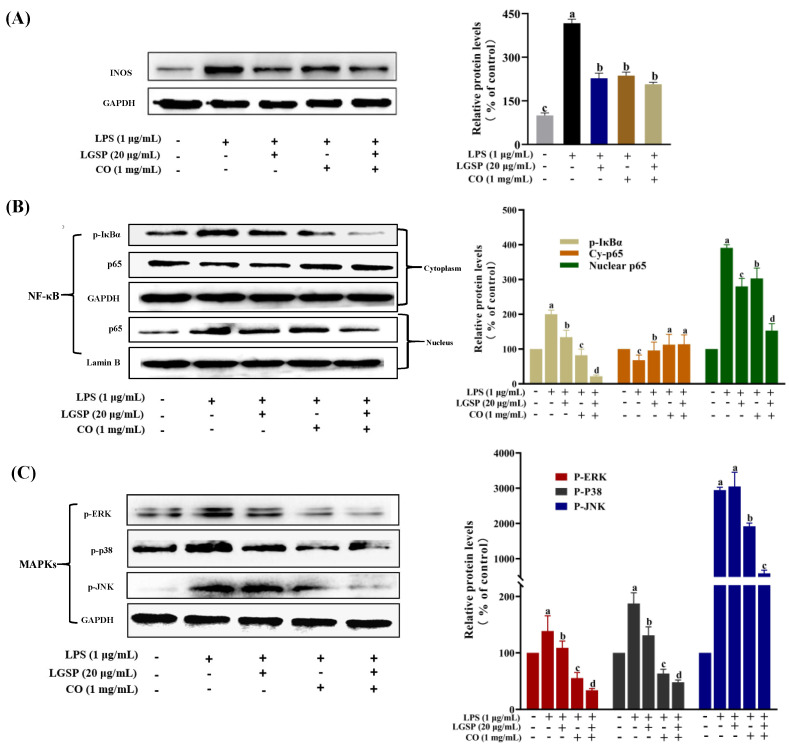
Effect of LGSP, CO and their combination on iNOS (**A**), NF-κB (**B**) and MAPKs (**C**) signaling pathways. All values are means ± SD, *n* = 3. Different letters above the bars indicate significant differences (*p* < 0.05).

**Figure 7 antioxidants-11-00289-f007:**
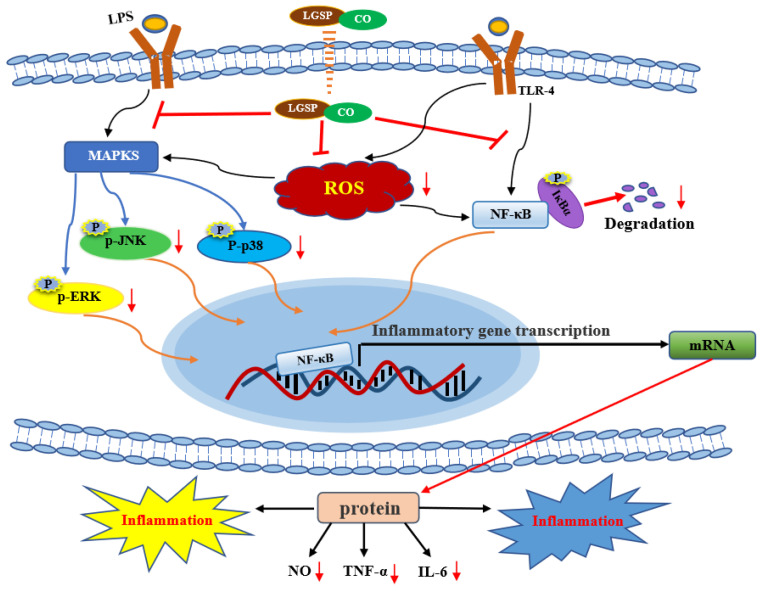
The underlying synergistic anti-inflammatory mechanism of LGSP and CO.

## Data Availability

Data is contained within the article and Appendix A.

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
