# Peer review of "Synergistic Anti-Inflammatory Effects of Lipophilic Grape Seed Proanthocyanidin and Camellia Oil Combination in LPS-Stimulated RAW264.7 Cells"

_antioxidants, 2022, doi:10.3390/antiox11020289_

Round 1

Reviewer 1 Report

In Fig. 2C, the legend and data do not match.

In Figure 6A, check the total form of IkBa and the band intensity and band pattern do not match.

Figure 6B Check the MAPK total form.

Unify the expression ex) RAW 264.7 vs RAW 264.7

Check the spacing between the numbers and the units ex) 45°C

Check the exponential expression ex) 1×104 cells/mL -> 104

Accurately indicate for statistical signatures, cannot determine exactly what a,b,c,d represents.

In Figure 4 (C-D), the decrease in IL-1beta and increase in IL-10 is not statistically sufficient, so it is difficult to indicate in Figure 7 that the regulation of IL-1beta and IL-10 is caused by the combination of LGSP and CO.

In Figure 4A, the increase in TNA alpha is not sufficient, so we will check again 

Introduction should include the roles and functions of cytokines identified in the above paper in the inflammatory response.

By checking the NO reduction pattern, authors will check the expression and activity patterns of genes or proteins that affect NO production.

It is necessary to check whether the combination of LGSP and CO has also antioxidant and anti-inflammatory effects in vivo.

In addition to DCFH-DA analysis, it is necessary to determine the antioxidant efficacy for LGSP and CO combination by various experiments such as DPPH, ABTS, and ORAC.

Reviewer 2 Report

The paper entitled "Synergistic anti-inflammatory effects of lipophilic grape seed proanthicyanidin and camelia oil combination in LPS-stimulated RAW264.7 cells" is a well written and a very interesting work. In my opinion there is almost nothing to correct or even to provide critical review. However, I would be pleased if the Authors could:

  1. provide the ED50 values for both compounds tested.
  2. provide detailed information on the solvent used in order to dissolve the compounds.
  3. the reason to use such concentrations of both drugs
  4. the subtitle 3.5. --> please provide the exact concentration of drugs' mixture used, as it is only written "combination". Similar information should be placed in the figure legend

Round 2

Reviewer 1 Report

In last review, the point 4 has not been revised.

Check the manucript carefully

ex) line81: RAW 264.7 -> RAW264.7

The answer to point 7 should be more clearly referred to the two groups in statistical significance

For example, statistical significance between the co-treatment group of LGSP&CO and the LGSP sole treatment group

The authors didn't find the significant differencen of  between expression of iNOS between their combination treatment and LGSP or CO individual treatment

Check the NO production according to the conditions that confirmed the iNOS protein level

Show these results (comparison results between combination treatment and individual treatment on NO production and iNOS protein level) and need to explain the results

Since this journal topics is antioxidant, it should show the results of synergistic effects of LSGP and CO in DPPH and ABTS as authors mentioned before in review process
